# Health problems, turnover intention, and actual turnover among shift work female nurses: Analyzing data from a prospective longitudinal study

**Jison Ki**[1], **Smi Choi-Kwon**[2]*

**1** The Research Institute of Nursing Science, Seoul National University, Seoul, Republic of Korea, **2** College of Nursing, The Research Institute of Nursing Science, Seoul National University, Seoul, Republic of Korea

* smi@snu.ac.kr

## Abstract

### Aims

This study investigated health problems, turnover intention, and actual turnover among shift work nurses. While turnover intention is often used as a proxy variable for turnover, the relationship between these variables requires clarification. This study tested for relevant associations using prospective longitudinal data with a time lag of 12 months. We also tested for associations between health problems (sleep disturbance, fatigue, and depression) and turnover intentions/turnover, with a focus on the mediating role of turnover intention.

### Methods

This study conducted a secondary analysis of data from the Shift Work Nurses' Health and Turnover project, which is a prospective longitudinal cohort study. We analyzed health problems, turnover intention, and actual turnover. The data were analyzed via descriptive statistics, the Pearson's chi-squared test, independent t-test, univariable logistic regression, multiple logistic regression, and causal mediation.

### Results

Participants included 491 shift work female nurses. Of these, 112 (22.8%) had turnover intention, while 38 (7.7%) left their jobs within the 12-month period of investigation. Of the 112 with turnover intention, 22 left their jobs (OR 5.68. 95% CI 2.84–11.36). The logistic regression analysis showed that sleep disturbance and fatigue were associated with turnover intentions and actual turnover, while depression was only associated with turnover intention. The causal mediation analysis showed that turnover intention mediated the relationship between health problems (sleep disturbance and fatigue) and actual turnover (sleep disturbance OR 1.31, 95%CI = 1.02–1.60; fatigue OR 2.11, 95%CI = 1.50–2.68); sleep disturbance had a natural direct effect on actual turnover (OR 2.66, 95%CI,2.07–3.21).

**Data Availability Statement:** All relevant data are within the paper and its Supporting Information file.

**Funding:** SC received funding. This work was supported by the National Research Foundation of Korea (NRF), grant funded by the Korea Ministry of Science, ICT & Future Planning (No. NRF-2019R1F1A1058862). The funders had no role in study design, data collection and analysis, decision to publish, or preparation of the manuscript.

**Competing interests:** The authors have declared that no competing interests exist.

## Conclusion

Turnover intention strongly predicted actual turnover. Sleep disturbances may result in turnover, even in cases without existing turnover intention. These findings highlight the need for early interventions aimed at preventing and alleviating sleep disturbances for shift work female nurses.

## Introduction

The high rate of turnover among nurses is a serious health system issue [1–3] that has been exacerbated by the COVID-19 pandemic [4]. Looking at conditions in Korea, the turnover rate has continually increased over the past few years, especially among novice nurses, who leave their jobs at triple the rate shown for all nurses nationwide [5].

The resulting nursing shortages have been associated with decreased patient satisfaction, an increased risk of infection, and longer hospital stays [6]. Meanwhile, nurses who experience high peer turnover are left with heavier workloads that can decrease job satisfaction [7]. This also poses issues related to job safety for nurses who continue to work during shortages, especially due to the increased risk of exposure to blood and other bodily fluids [8]. From the administrative standpoint, hospitals are faced with additional financial burdens due to the loss of skilled nurses and subsequent need to recruit and train new nurses [9].

In previous research, turnover intention has been used as a proxy variable for actual turnover, as it may be the best predictor according to the Theory of Planned Behavior [10, 11]. This is also because actual turnover is difficult to investigate among nurses, particularly due to the need for a longitudinal approach, which requires more time, financing, and effort than cross-sectional research. Further, ethical issues may arise when tracking these behaviors through personal information, which is often required in longitudinal surveys [12]. Amid these concerns, there is a general lack of research on the relationship between turnover intention and actual turnover, with inconsistent results between studies [2, 10]. Finally, reports have shown that health problems such as sleep disturbances, fatigue, and depression can influence turnover intention among nurses, but there is a lack of evidence about the relationships between these health problems and actual turnover [13–15].

Due to these gaps in the literature, this study examined the relationships between health problems, turnover intention, and actual turnover by conducting a secondary analysis of longitudinal data with a time lag of 12 months. To do so, we explored the associations between each health problem and turnover intention/actual turnover, with a focus on the mediating role of turnover intention in the relationship between health problems and actual turnover.

## Materials and methods

### Study design

This study conducted a secondary analysis of cohort data collected via the Shift Work Nurses' Health and Turnover (SWNHT) project (2018–2020).

### Data source and sampling

The SWNHT was a longitudinal prospective cohort study that investigated relationships between health and turnover among shift work nurses in Korea. The detailed study methods have been published [13]. Participants included 594 female nurses (i.e., 294 novice nurses with

no exposure to rotating shift work and 300 nurses with exposure to eight-hour rotational work, including night shifts, for a period lasting at least one month). Because health problems may vary according to sex [16, 17], the SWNHT was limited to female nurses. For novice nurses, data were collected three times, including before exposure to shift work (novice registered nurse [NRN] T0, n = 294), six months after work (NRN T1, n = 204), and 12 months after T1 (NRN T2, n = 204). For experienced registered nurses, data were collected twice, including at baseline (experienced registered nurse [ERN] T1, n = 300) and 12 months after T1 (ERN T2, n = 269). To enroll nurses, we attached a recruitment notice to the ward bulletin boards and also distributed survey envelope packages after their lecture time in the hospital. Nurses who wished to participate in the study voluntarily contacted the research team. After agreeing to participate in the study, all nurses signed the consent form and completed the baseline questionnaire. All data were collected between March 2018 and April 2020 at two tertiary hospitals in Seoul, South Korea. The number of shift work nurses in the two study hospitals was approximately 1,400 and 1,900, and data from 244 and 247 nurses were analyzed for this study, respectively. Records of nurses leaving the hospital were surveyed from the participants or the nursing department in which they worked during the T2 survey. The SWNHT was approved by the institutional review boards at both tertiary hospitals (IRB No. H-1712-094-907, 2017-12-075-002). To secondary analysis data from the SWNHT, this study was approved by the institutional review boards at Seoul National University (IRB No. E2011/003-012).

In this study, we analyzed data from both NRN T1 (n = 204) and ERN T1 (n = 300), thus spanning a collection period lasting from March 2018 to January 2019. We defined shift work as a combination of day, evening, and night shifts with varying numbers of shifts in a row of 1–5 days and included shift work female nurses with at least 1 month of experience and no omission in major variables. According to these criteria, this subset included a final sample size of 491 participants, excluding 12 with no engagement in eight-hour rotational shift work and one who did not answer items related to the major variables. For these participants, actual turnover was checked at T2, 12 months after the NRN T1, and ERN T1 surveys.

## Measures

**General characteristics.**    Each participant provided their information for the following: age (years), education level (bachelor's degree or lower/master's degree or higher), marital status (single/married), whether they had children (yes/no), work unit (general ward, intensive care unit, delivery room, or emergency room), and total shift work experience (less than one year/more than one year).

**Turnover intention and actual turnover.**    Turnover intention is defined as an employee's voluntary resignation intention or attempt [18, 19]. In our study, each participant answered the following question using one of four options (strongly agree, agree, disagree, or strongly disagree): "I plan on staying for the next year" [20]. Prior to the analysis, turnover intention was converted into a dichotomous mediating variable, in which 0 = intent to stay (strongly agree or agree) and 1 = intent to leave (disagree or strongly disagree). Based on turnover data from the T2 survey, we determined whether actual turnover had occurred within a 12-month period following the completion of the T1 survey.

**Sleep disturbance.**    Sleep disturbance was measured using the Korean version of the Insomnia Severity Index (ISI), which was developed by Morin [21] and translated by the Korean Sleep Research Society [22]. The Korean ISI comprised of seven items that are rated in a 5-point scale (0–4 points), and the score ranges from 0 to 28. Higher scores indicate lower sleep quality and scores above 10 indicate sleep disturbance during the past 2 weeks [21]. In the study context, the Korean ISI received a Cronbach's alpha of 0.92.

**Fatigue.**   Fatigue was measured using the Fatigue Severity Scale (FSS), which comprised of nine items concerning the degree of fatigue over the previous week. Each item is rated in a 7-point Likert scale (1 = strongly disagree, 7 = strongly agree), and average scores obtained by dividing the total score (range 9–63) by the number of items indicates the degree of fatigue; the cutoff point for fatigue is more than 4 points on average [23]. In the study context, the FSS received a Cronbach's alpha of 0.91.

**Depression.**   Depression was assessed using the shortened Center for Epidemiological Studies Depression Scale (CES-D), which is comprised of 10 items concerning depressive feelings and thoughts during the previous week. Each item is rated in a 4-point scale (0 = less than one day, 3 = about five to seven days), with higher total scores (range 0–30) indicating more depressive symptoms; total scores of 10 or above indicate depression [24]. In the study context, the shortened CES-D received a Cronbach's alpha of 0.87.

## Statistical analyses

All analyses were conducted using SAS version 9.4 (SAS Institute Inc., Cary, NC, USA). Descriptive statistics (frequencies, percentages, means, and standard deviations) were analyzed for the general characteristics, while Pearson's chi-squared test and independent t-test were used to identify the differences between turnover intention and actual turnover according to the general characteristics and health problems.

Multiple logistic regressions were used to investigate the associations between health problems and turnover intention, including all general characteristics except for age used as covariates, as age was highly correlated with total shift work experience ($r = 0.92$, $p < 0.001$). In the multiple logistic regressions for actual turnover, we added turnover intention to the models to confirm the weakening associations between health problems and actual turnover.

We conducted a causal mediation analysis (CAUSALMED procedure in SAS) to explore the mediating effects of turnover intention. Specifically, the causal mediation analysis is a statistical method based on the counterfactual framework, and which enables researchers to estimate exactly even when the mediator and outcome are dichotomous or if there is an interaction between the exposure and the mediator [25]. Three estimates were obtained through the analysis, including the total effect (TE), natural direct effect (NDE), and natural indirect effect (NIE). TE is the sum of the NDE and NIE, while NDE is the effect of a shift in the outcome based on exposure, assuming that the mediator is fixed, and NIE is the effect of a shift in the outcome based on the mediator, assuming that the exposure is fixed. We also obtained the mediated percentage, which is the proportion of NIE of TE [26]. Fig 1 shows a directed acyclic graph of this causal mediation analysis.

## Results

### Participant characteristics

As mentioned, the analyzed participants included 491 female nurses with exposure to shift work, including night shifts. Their mean age was 26.1 years (standard deviation [SD] = 4.30); 87.8% (n = 431) were single. Most had less than five years of experience as nurses (n = 384, 78.2%). Of all participants, 112 (22.8%) had turnover intention and 38 (7.7%) left their jobs. Of the 112 (22.8%) with turnover intention, 22 left their jobs (OR, 5.54. 95% CI 2.79–10.99). Notably, the prevalence of both sleep disturbance and fatigue was significantly higher among the participants with turnover intention and those who left their jobs. Meanwhile, the prevalence of depression was significantly higher among those with turnover intention ($\chi^2 = 15.34$, p<0.001) (Table 1).

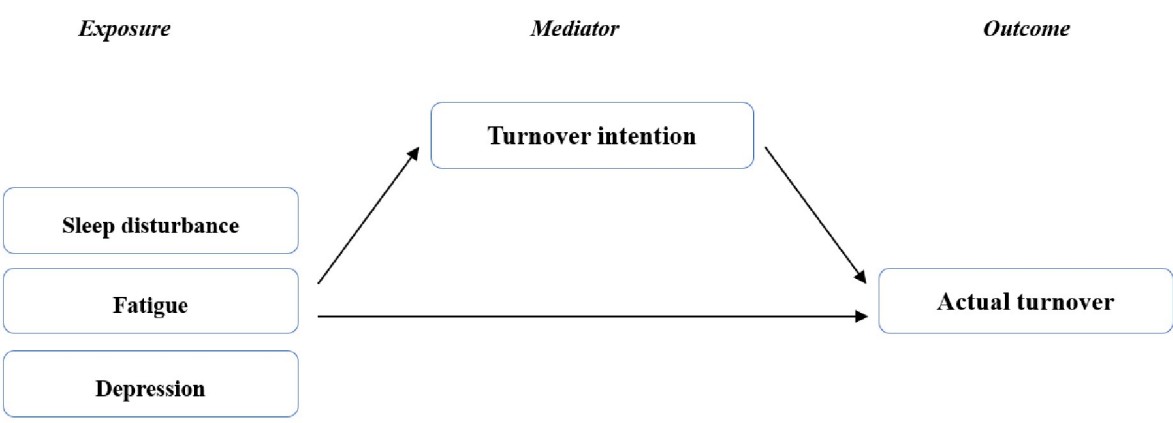

**Fig 1. Directed acyclic graph for the mediation analysis.**

## Association between health problems and turnover intention/actual turnover

According to the multiple logistic regression analysis with covariates, sleep disturbance, fatigue, and depression were each associated with turnover intention (Table 2). Moreover,

**Table 1. General characteristics and health problems by turnover intention and actual turnover.**

| Variables | Categories | Total | Intend to stay | Intend to leave | $\chi^2$ or t | P | Stayer | Leaver | $\chi^2$ or t | P |
|---|---|---|---|---|---|---|---|---|---|---|
| | | (n = 491, 100.0%) | (n = 379, 77.2%) | (n = 112, 22.8%) | | | (n = 453, 92.3%) | (n = 38, 7.7%) | | |
| | | n(%) or M±SD | n(%) or M±SD | n(%) or M±SD | | | n(%) or M±SD | n(%) or M±SD | | |
| Age (years) | | 26.1±4.30 | 26.00±4.06 | 26.44±5.02 | -0.85 | 0.399 | 26.15±4.43 | 25.52±2.12 | 1.56 | 0.123 |
| Education | ≤ BSN | 449(91.5) | 348(91.8) | 101(90.2) | 0.30 | 0.585 | 412(91.0) | 37(97.4) | 1.85 | 0.235 |
| | ≥ MSN | 42(8.5) | 31(8.2) | 11(9.8) | | | 41(9.0) | 1(2.6) | | |
| Marital status | Single | 431(87.8) | 335(88.4) | 96(85.7) | 0.58 | 0.447 | 395(87.2) | 36(94.7) | 1.86 | 0.173 |
| | Married | 60(12.2) | 44(11.6) | 16(14.3) | | | 58(12.8) | 2(5.3) | | |
| Has children | Yes | 31(6.3) | 23(6.1) | 8(7.1) | 0.17 | 0.681 | 31(6.8) | 0(0.0) | 2.78 | 0.096 |
| | No | 460(93.7) | 356(93.9) | 104(92.9) | | | 422(93.2) | 38(100.0) | | |
| Work unit | Ward | 364(74.1) | 278(73.4) | 86(76.8) | 0.62 | 0.733 | 333(73.5) | 31(81.6) | 1.19 | 0.550 |
| | ICU | 110(22.4) | 87(22.9) | 23(20.5) | | | 104(23.0) | 6(15.8) | | |
| | DR, ER | 17(3.5) | 14(3.7) | 3(2.7) | | | 16(3.5) | 1(2.6) | | |
| Total shift work experience | Less than one year | 233(47.5) | 186(49.1) | 47(42.0) | 1.75 | 0.185 | 215(47.5) | 18(47.4) | 0.00 | 0.991 |
| | One year or more | 258(52.5) | 193(50.9) | 65(58.0) | | | 238(52.5) | 20(52.6) | | |
| Sleep disturbance | Yes | 307(62.5) | 220(58.1) | 87(77.7) | 14.22 | <0.001** | 275(60.7) | 32(84.2) | 8.27 | 0.004** |
| | No | 184(37.5) | 159(41.9) | 25(22.3) | | | 178(39.3) | 6(15.8) | | |
| Fatigue | Yes | 321(65.4) | 227(59.9) | 94(83.9) | 22.06 | <0.001** | 289(63.8) | 32(84.2) | 6.45 | 0.011* |
| | No | 170(34.6) | 152(40.1) | 18(16.1) | | | 164(36.2) | 6(15.8) | | |
| Depression | Yes | 202(41.1) | 138(36.4) | 64(57.1) | 15.34 | <0.001** | 183(40.4) | 19(50.0) | 1.34 | 0.248 |
| | No | 289(58.9) | 241(63.6) | 48(42.9) | | | 270(59.6) | 19(50.0) | | |

BSN = Bachelor of Science in Nursing; MSN = Master of Science in Nursing; ICU = intensive care unit; DR = delivery room; ER = emergency room

*$p<0.05$

**$P<0.01$

**Table 2. Associations between health problems and turnover intention according to the multiple logistic model.**

| Variable | Between sleep disturbance and turnover intention | | Between fatigue and turnover intention | | Between depression and turnover intention | |
|---|---|---|---|---|---|---|
| | OR | 95% CI | OR | 95% CI | OR | 95% CI |
| Sleep disturbance | 2.50** | 1.52–4.11 | NA | | NA | |
| Fatigue | NA | | 3.52** | 2.03–6.10 | NA | |
| Depression | NA | | NA | | 2.70** | 1.72–4.24 |

Adjusted for education, marital status, children (yes/no), work unit, and total shift work experience

*p<0.05

**P<0.01

turnover intention was significantly associated with actual turnover (OR 5.68, 95%CI = 2.84–11.36). While sleep disturbance was associated with actual turnover regardless of whether we adjusted for turnover intention (OR 3.48, 95%CI = 1.41–8.57), this association was weakened by the inclusion of turnover intention (OR 2.73, 95%CI = 1.08–6.83). Finally, fatigue was positively associated with actual turnover (OR 3.03, 95%CI = 1.23–7.45), but this was no longer significant after adjusting for turnover intention (Table 3).

## Mediating effects of turnover intention

Sleep disturbance and fatigue had significant effects on actual turnover (sleep disturbance OR 3.50, 95%CI = 2.15–4.80; fatigue OR 3.23, 95%CI = 1.02–5.01), while turnover intention mediated sleep disturbance, fatigue, and depression (NIE of sleep disturbance OR 1.31, 95% CI = 1.02–1.60; NIE of fatigue OR 2.11, 95%CI = 1.50–2.68; NIE of depression OR 1.34, 95% CI = 1.02–1.72). The NDE of sleep disturbance was also significant (OR 2.66, 95%CI,2.07–3.21, Table 4).

## Discussion

This study investigated the longitudinal relationships between health problems, turnover intention, and actual turnover among shift work nurses. First, we found that turnover intention significantly influenced actual turnover. Second, the participants frequently experienced a variety of health problems that affected their work. Here, the analyses showed that sleep disturbance and fatigue were each associated with turnover intention and turnover, while depression

**Table 3. Associations between health problems and actual turnover according to the multiple logistic model.**

| Variable | Between turnover intention and actual turnover | | Between sleep disturbance and actual turnover | | Among sleep disturbance, turnover intention, and actual turnover | | Between fatigue and actual turnover | | Among fatigue, turnover intention, and actual turnover | | Between depression and actual turnover | | Among depression, turnover intention, and actual turnover | |
|---|---|---|---|---|---|---|---|---|---|---|---|---|---|---|
| | OR | 95% CI | OR | 95% CI | OR | 95% CI | OR | 95% CI | OR | 95% CI | OR | 95% CI | OR | 95% CI |
| Turnover intention | 5.68** | 2.84–11.36 | NA | | 5.02** | 2.48–10.13 | NA | | 4.94** | 2.43–10.03 | NA | | 5.72** | 2.79–11.68 |
| Sleep disturbance | NA | | 3.48** | 1.41–8.57 | 2.73* | 1.08–6.83 | NA | | NA | | NA | | NA | |
| Fatigue | NA | | NA | | NA | | 3.03* | 1.23–7.45 | 2.11 | 0.83–5.37 | NA | | NA | |
| Depression | NA | | NA | | NA | | NA | | NA | | 1.42 | 0.71–2.81 | 0.97 | 0.46–2.02 |

Adjusted for education, marital status, children (yes/no), work unit, and total shift work experience

*p<0.05

**P<0.01

**Table 4. The mediating effect of turnover intention in the relationship between health problems and actual turnover.**

| Mediation of turnover intention | Association of sleep disturbance and actual turnover | | Association of fatigue and actual turnover | | Association of depression and actual turnover | |
|---|---|---|---|---|---|---|
| | OR | 95% CI | OR | 95% CI | OR | 95% CI |
| Total effect | 3.50** | 2.15–4.80 | 3.23* | 1.02–5.01 | 1.35 | 0.33–2.37 |
| Natural direct effect | 2.66** | 2.07–3.21 | 1.52 | 0.59–2.46 | 1.01 | 0.28–1.74 |
| Natural indirect effect | 1.31* | 1.02–1.60 | 2.11** | 1.50–2.68 | 1.34* | 1.02–1.72 |
| Percentage mediated, % | 33.5 | | 50.2 | | 98.0 | |

Adjusted for education, marital status, children (yes/no), work unit, and total shift work experience (causal mediation analysis)

*$p<0.05$

**$P<0.01$

was only associated with turnover intention. Surprisingly, we also found that sleep disturbance had a direct effect on actual turnover, which indicates that participants were prone to abruptly leaving their nursing jobs upon suffering from sleep disturbance, even in cases without existing turnover intention.

Shift work can cause a variety of physical and mental health problems and could become more serious as the period increases [27]. About 75% of Korean nurses work in 8-hr rotations including day, evening, and night shifts [28], which means that Korean nurses' health may be vulnerable. In previous studies, it has been reported that nurses' health problem not only lower the quality of life but also affect the quality of nursing, which in turn can lead to turnover intention [29, 30]. In this study, 22.8% of participants had turnover intention; of this subset, 19.6% ultimately left their jobs within a 12-month period. Meanwhile, the literature shows a variety of different turnover intention rates (4~64%) in various nursing samples [31–35]. Such a wide range may partially be due to the different measurement tools used between studies. While this study considered turnover intention based on responses to items such as "I plan on staying for the next year", other studies used different reference points, including how often participants thought about turnover in the past [35] and whether they were looking for other jobs [34]. The actual reported turnover rates (2~63%) also differ between studies, which may be due to the various definitions used for turnover [35–39]. While this study defined turnover as an event in which a given participant terminated their employment by leaving the hospital, other studies defined turnover as an event in which the participant transferred to a different work unit or completely left the nursing sector [35, 36]. Above all, such differences may be due to the wide diversity of hospital environments. We recruited participants who worked in tertiary hospitals with a relatively high nurse-to-patient ratio. Most of the participants worked in general wards or intensive care units, taking care of an average of 12 patients in general wards and 2–3 patients in intensive care units. While the severity of patients was high, the salary and working conditions were better than it is in other hospitals in Korea. However, other studies have reported turnover rates in different hospital environments in various countries, regions, and medical institutions [39].

We found a high causality between turnover intention and turnover (OR, 5.54. 95% CI 2.79–10.99). Our causal mediation analysis further revealed that the associations between health problems and actual turnover were partially mediated by turnover intention, meaning that sleep disturbance and/or fatigue may lead to actual turnover by increasing turnover intention. These results support previous studies showing that turnover intention plays mediating roles in the relationships between job satisfaction and actual turnover, quality of work life and actual turnover [36–38].

Among the health problems that were related to actual turnover, sleep disturbance had the greatest total effect. Surprisingly, the natural direct effect on actual turnover was also statistically significant, which indicates that sleep disturbance may affect actual turnover without increasing turnover intention. Of the 16 nurses who did not have previous turnover intention but ultimately left their jobs, 14 experienced sleep disturbance. This shows that poor sleep quality may induce other problems that influence voluntary job termination. Indeed, previous studies have shown that sleep disturbance affects nursing performance by impairing cognitive function, empathy, and judgment, and increased the potential for errors that threaten patient safety [40–43]. Further, these conditions pose risks that may impact physical and mental health for nurses, including gastrointestinal disorders, cardiovascular disease, obesity, diabetes, and depression [27, 41, 44].

Previous studies have reported a prevalence of sleep disturbance ranging from 57–83% in shift work nurses, which is higher than that found for the general adult population [44, 45]. In this study, a significant number of participants (62%) had sleep disturbance, with an increased prevalence among those with turnover intention (77.7%) and those who left their jobs (84.2%). A number of previous studies have similarly reported associations between sleep disturbance and turnover [14, 15]. As a whole, these findings show that sleep disturbance is a serious health problem for shift work nurses, thus emphasizing the need for multi-faceted organizational support.

In addition, we found that fatigue may cause actual turnover by increasing turnover intention. However, fatigue has received relatively less attention than sleep disturbance due to the known associations between turnover intention and this latter variable [46]. Still, fatigue is a common symptom for shift workers [47, 48]. In this study, 65.4% of all participants and 84.2% of those who left their jobs reported fatigue, which has previously been associated with an increased potential for chronic disease and mental stress [46, 49]. In nurses, fatigue disrupts optimal performance and negatively affects skills while also increasing the risks for medical error, exposure to blood and other body fluids, and musculoskeletal disorder [50]. It is noteworthy that the current results suggest that fatigue may affect both turnover intention and actual turnover. In this regard, managers should work to identify fatigue-related factors and possibly adopt flexible strategies to reduce fatigue among shift work nurses, as this may promote health while reducing actual turnover.

In a previous cross-sectional study, depression was positively associated with turnover intention among nurses [13, 51]; however, a longitudinal cohort study found no such association with actual turnover [52]. Likewise, this study found that depression was associated with turnover intention but not with actual turnover. However, the prevalence of depression is lower in our study than as reported in previous research [53]. We also did not investigate whether participants were taking antidepressants or had been diagnosed with depression. These should be considered when interpreting the results.

Although this study produced new evidence concerning the relationships between health problems, turnover intention, and actual turnover, there were also some limitations. First, we only considered the most commonly reported health problems in shift workers. These health problems could have many other causes besides shift work and may have preceded nurses' exposure to shift work. In addition, other untested health issues may have had affected turnover intention/turnover. Second, because there is a high turnover rate among novice nurses in Korea, the SWNHT included a relatively high proportion of such participants. In this regard, results may differ when including a higher proportion of experienced nurses. Third, the follow-up period was only 12 months. Additional research is needed to investigate nurse turnover and any related factors over longer periods of time.

## Conclusions

In this study, turnover intention was a strong predictor of actual turnover. This highlights the potential connections between turnover intention and actual turnover. Further, turnover intentions led to turnover after some period of time, meaning that turnover was not a sudden event, but a multi-stage process. Finally, there is s strong potential that sleep disturbance may cause turnover, even for nurses without existing turnover intention. This urgently emphasizes the need for early interventions aimed at preventing and alleviating sleep disturbances among shift work nurses.

## Supporting information

**S1 File.**
(SAS7BDAT)

## Author Contributions

**Conceptualization:** Jison Ki, Smi Choi-Kwon.

**Data curation:** Jison Ki, Smi Choi-Kwon.

**Formal analysis:** Jison Ki.

**Funding acquisition:** Smi Choi-Kwon.

**Methodology:** Jison Ki.

**Supervision:** Smi Choi-Kwon.

**Writing – original draft:** Jison Ki.

**Writing – review & editing:** Smi Choi-Kwon.

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
