## [Decision Letter · Decision Letter 0]

7 Mar 2022

PONE-D-21-32791Health problems, turnover intention, and actual turnover among shift work nurses: Analyzing data from a prospective longitudinal studyPLOS ONE

Dear Dr. Choi-Kwon,

Thank you for submitting your manuscript to PLOS ONE. After careful consideration, we feel that it has merit but does not fully meet PLOS ONE’s publication criteria as it currently stands. Therefore, we invite you to submit a revised version of the manuscript that addresses the points raised during the review process.

We look forward to receiving your revised manuscript.

Kind regards,

Mohammad Hossein Ebrahimi

Academic Editor

PLOS ONE

Journal Requirements:

2. Please provide additional details regarding participant consent. In the ethics statement in the Methods and online submission information, please ensure that you have specified (1) whether consent was informed and (2) what type you obtained (for instance, written or verbal, and if verbal, how it was documented and witnessed). If the need for consent was waived by the ethics committee, please include this information.

**Comments to the Author**

1. Is the manuscript technically sound, and do the data support the conclusions?

Reviewer #1: Partly

Reviewer #2: Partly

Reviewer #3: Yes

2. Has the statistical analysis been performed appropriately and rigorously? 

Reviewer #1: Yes

Reviewer #2: Yes

Reviewer #3: Yes

3. Have the authors made all data underlying the findings in their manuscript fully available?

Reviewer #1: Yes

Reviewer #2: Yes

Reviewer #3: No

4. Is the manuscript presented in an intelligible fashion and written in standard English?

Reviewer #1: Yes

Reviewer #2: Yes

Reviewer #3: Yes

5. Review Comments to the Author

Reviewer #1: The authors explored the association between health problems and turnover intention/actual turnover, approaching sleep disturbance, depression and fatigue as health problems, using validated instruments. Sample consists of young women, which may limit the extrapolation of results. However, some comments are needed:

1- What was considered shift work? Was this work organization uniform among the participants?

2- Participants were classified in relation to the total shift work experience as less than one year/more than one year, which can be considered a short time in shift work, especially in relation to the chronic effects on the health of health care workers (HCWs) . What is the reason for this classification? What is the impact of this difference on the analysis of results?

3- Was the variable considered in the analysis for another job concurrently with the current one? Is this information available?

4- Regarding the instruments used, what is the previous period considered in the sleep disturbance and depression analyzes (previous week, previous month)?

5- Shift work can impact the health of HCWs, including aggravated by the time of exposure to risk. In this sense, other health problems that may also impact the turnover intention/actual turnover were evaluated, for example: cardiovascular, metabolic, musculoskeletal diseases?

Reviewer #2: This is quite interesting work about the relationship between shift work and some health problems among nurses. It is well written and quite well structured but, in my opinion, it lacks of numerous basic data. There are many points that need to be better clarified.

Which is the the reason why, for novice nurses, data were collected 3 times instead of 2.

It is known that the health problems explored (fatigue, depression, sleep disturbance) may have multiple causes, why they haven't taken into consideration? Moreover, there is no information about the previous presence of those problems among the population studied.

The definition applied for "turnover intention" should be indicated first in the text.

There is no information about the hospital ward where the participants work. I think that this information should be very useful to better understand the reasons for turnover intention.

It is not discussed the relationship between shift work experience and turnover intention, even if the study sample was mainly made of shift workers.

The inclusion criteria of the respondents have not been indicated, even if referred to a previous study, it should be useful to report them in the methods section.

Reviewer #3: Dear authors thank you for this research:

1- I suggest adding female nurses in the title as you only include them and as you mention the turnover intention differ between both genders.

2- could you give us some few details about the number of hospitals from which the nurses were included

3- write the significance level under the tables or the statistical analysis part of the methods.

4- for the questionnaire used please add the total score range for each domain you examine( according to the number of questions and the score points).

5- was a pilot study conducted to assess the clarity of the questionnaires

Good luck

6. PLOS authors have the option to publish the peer review history of their article (what does this mean?). If published, this will include your full peer review and any attached files.

Reviewer #1: No

Reviewer #2: **Yes: **Giuseppe Buomprisco

Reviewer #3: No

---

## [Author Response · Author response to Decision Letter 0]

20 May 2022

Reviewer #1: The authors explored the association between health problems and turnover intention/actual turnover, approaching sleep disturbance, depression and fatigue as health problems, using validated instruments. Sample consists of young women, which may limit the extrapolation of results. However, some comments are needed:

Thank you for your valuable feedback. Our responses to your remarks are presented in bold.

1. What was considered shift work? Was this work organization uniform among the participants?

Thank you for pointing this out. We defined shift work as a combination of 8-hour day, evening, and night shifts with varying numbers of shifts in a row of 1–5 days. All participants in our study were exposed to the same pattern of shift work. We have now added our definition of shift work and inclusion criteria of participants in the Materials and Methods section. (page 6, lines 109-111).

“We defined shift work as a combination of day, evening, and night shifts with varying numbers of shifts in a row of 1–5 days and included shift work female nurses with at least 1 month of experience and no omission in major variables.”

2. Participants were classified in relation to the total shift work experience as less than one year/more than one year, which can be considered a short time in shift work, especially in relation to the chronic effects on the health of health care workers (HCWs) . What is the reason for this classification? What is the impact of this difference on the analysis of results?

Thank you for pointing this out. There are three reasons for classifying the total shift work experience as less than one year/more than one year in our study. First, a previous study reported that the competency of novice nurses improved around 1 year after they started working as nurses, and suggested 1 year as a period to distinguish between novice nurses and experienced nurses . Second, the variables (fatigue, depression, and sleep) we considered in this study are known as chronic health problems in shift workers, but they are also symptoms that appear highest around 6 months after starting shift work. After the period, these health problems either slightly improve or remain over time . Lastly, the turnover rate of nurses in the first year was the highest in Korea. Since the turnover rate was one of the important variables in our study, we thought it was reasonable to divide the subjects into two groups (less than one year/more than one year) considering the aforementioned results.

3. Was the variable considered in the analysis for another job concurrently with the current one? Is this information available? 

Thank you for pointing out a very important issue. Although we did not study another job, a previous study reported that shift work was associated with long sleep and increased fatigue in hospital workers with various job titles including nurses . In addition, there was a study that reported that sleep problems and mental disorders are caused by shift work in various occupations, using national data . In a future study, if we consider the same variables in another job, participants’ characteristics in different occupations could be well represented.

4. Regarding the instruments used, what is the previous period considered in the sleep disturbance and depression analyzes (previous week, previous month)?

We apologize for not clearly describing the period. The period suggested in the instruments is the last 2 weeks for sleep disturbance and 1 week for fatigue and depression. We added information about the period specified in instruments in the Materials and Methods section. (page 7, lines 138-139, lines 143-144, page 8, lines 151-153).

“Higher scores indicate lower sleep quality and scores above 10 indicate sleep disturbance during the past 2 weeks [21].”

“Fatigue was measured using the Fatigue Severity Scale (FSS), which comprised of nine items concerning the degree of fatigue over the previous week.”

“Depression was assessed using the shortened Center for Epidemiological Studies Depression Scale (CES-D), which comprised of 10 items concerning depressive feelings and thoughts during the previous week.”

5. Shift work can impact the health of HCWs, including aggravated by the time of exposure to risk. In this sense, other health problems that may also impact the turnover intention/actual turnover were evaluated, for example: cardiovascular, metabolic, musculoskeletal diseases?

Unfortunately, we did not consider a cardiovascular or metabolic disease. The participants of our study were young female nurses with an average age of 26 which also corresponds to those of Korean average female nurses. Considering these conditions, it was expected that the prevalence of cardiovascular or metabolic diseases was very low and that the association with turnover could not be explored. We previously reported, however, that there was no association with turnover intention in musculoskeletal diseases .

Reviewer #2: This is quite interesting work about the relationship between shift work and some health problems among nurses. It is well written and quite well structured but, in my opinion, it lacks of numerous basic data. There are many points that need to be better clarified.

Thank you for your valuable feedback. Our responses to your remarks are presented in bold.

1. Which is the reason why, for novice nurses, data were collected 3 times instead of 2.

Thank you for your interest in our study. The reason we conducted the survey three times for novice nurses is that, unlike experienced nurses, one additional survey was conducted to explore the health status before exposure to shift work, which was excluded on the part of the experienced nurses. The other two surveys were conducted after six months with a time lag of 12 months for novice nurses, which is similar to that of experienced nurses (Figure 1). 

Figure 1. Schematic overview of the Shift Work Nurses’ Health and Turnover (SWNHT) study. Reprinted from “Association between Health Problems and Turnover Intention in Shift Work Nurses: Health Problem Clustering”, by J-S. Ki et al., 2020, Int. J. Environ. Res. Public Health, 17(12), 4532.

2. It is known that the health problems explored (fatigue, depression, sleep disturbance) may have multiple causes, why they haven't taken into consideration? Moreover, there is no information about the previous presence of those problems among the population studied.

Thank you for pointing out a very important issue. We agree with the reviewer. We assumed that the health problems explored in our study (sleep disturbance, fatigue, and depression) were caused mainly by shift work. However, as you pointed out, these health problems could have many other causes besides shift work and some of these problems could have preceded the shift nurses’ exposure to shift work. We now have added this as a limitation in the Discussion section as the reviewer suggested (page 18, lines 310-314).

“First, we only considered the most commonly reported health problems in shift workers. These health problems may have many other causes besides shift work and may have preceded nurses’ exposure to shift work. In addition, other untested health issues may have had affected turnover intention/turnover.”

3. The definition applied for "turnover intention" should be indicated first in the text.

 We apologize for the omission of the definition. We have added the definition of the turnover intention in the Materials and Methods section (page 7, lines 125-126).

“Turnover intention is defined as an employee's voluntary resignation intention or attempt[18, 19].”

4. There is no information about the hospital ward where the participants work. I think that this information should be very useful to better understand the reasons for turnover intention.

Thank you for your suggestion. We added information about the hospitals where the participants work in the Discussion section (page 16, lines 257-263 ). 

“We recruited participants who worked in tertiary hospitals with a relatively high nurse-to-patient ratio. Most of the participants worked in general wards or intensive care units, which took care of an average of 12 patients in general wards and 2-3 patients in intensive care units. While the severity of patients was high, the salary and working conditions were superior to other hospitals in Korea. However, other studies have reported turnover rates in different hospital environments in various countries, regions, and medical institutions [39].”

5. It is not discussed the relationship between shift work experience and turnover intention, even if the study sample was mainly made of shift workers.

Thank you for your suggestion. We have now added the relationship between shift work and turnover intention in the Discussion section (page 15, lines 240-244 ).

“Shift work can cause a variety of physical and mental health problems and could become more serious as the period increases [27]. About 75% of Korean nurses work in 8‐hr rotations including day, evening, and night shifts[28], which means that Korean nurses' health may be vulnerable. In previous studies, it has been reported that nurses’ health problems not only lower the quality of life but also affect the quality of nursing, which in turn can lead to turnover intention [29, 30].”

6. The inclusion criteria of the respondents have not been indicated, even if referred to a previous study, it should be useful to report them in the methods section.

Thank you for pointing this out. we have now added our definition of shift work and inclusion criteria of participants in the Materials and Methods section. (page 6, lines 109-114).

“We defined shift work as a combination of day, evening, and night shifts with varying numbers of shifts in a row of 1–5 days and included shift work female nurses with at least 1 month of experience and no omission in major variables. According to these criteria, this subset included a final sample size of 491 participants, excluding 12 with no engagement in eight-hour rotational shift work and one who did not answer items related to the major variables.”

Reviewer #3: Dear authors thank you for this research:

Thank you for your valuable feedback. Our responses to your remarks are presented in bold.

1. I suggest adding female nurses in the title as you only include them and as you mention the turnover intention differ between both genders.

Thank you for pointing this out. We agree with the reviewer’s opinion. We have changed the title to “Health problems, turnover intention, and actual turnover among shift work female nurses: Analyzing data from a prospective longitudinal study”.

2. could you give us some few details about the number of hospitals from which the nurses were included

All data were collected at two tertiary hospitals in Seoul, South Korea. The number of shift work nurses in the two study hospitals averaged 1,700 with an average of 1,800 beds. For this study, we analyzed data from 244 and 247 nurses, respectively. We have added these details in the Materials and Methods section (page 6, lines 101-103)

“The number of shift work nurses in the two study hospitals was approximately 1,400 and 1,900, and data from 244 and 247 nurses were analyzed for this study, respectively.”

3. write the significance level under the tables or the statistical analysis part of the methods.

Thank you for your suggestion. We have now added the significance level under the tables (Table 1, Table 2, Table 3, Table 4).

4. for the questionnaire used please add the total score range for each domain you examine (according to the number of questions and the score points).

Thank you. We have added the total score range for the key variables.

Sleep Disturbance (page 7, lines 137-139)

“The Korean ISI comprised of seven items that are rated in a 5-point scale (0–4 points), and the score ranges from 0 to 28. Higher scores indicate lower sleep quality and scores above 10 indicate sleep disturbance during the past 2 weeks [21].”

Fatigue (page 7, lines 144-147)

“Each item is rated in a 7-point Likert scale (1 = strongly disagree, 7 = strongly agree), and average scores obtained by dividing the total score(range 9-63) by the number of items indicates the degree of fatigue; the cutoff point for fatigue is more than 4 points on average [23].”

Depression (page 8, lines 153-155)

“Each item is rated in a 4-point scale (0 = less than one day, 3 = about five to seven days), with higher total scores (range 0-30) indicating more depressive symptoms; total scores of 10 or above indicate depression [24].”

5. was a pilot study conducted to assess the clarity of the questionnaires

Yes, you are right. We previously conducted a pilot study and published it. 

(Baek J, Choi-Kwon S. Sleep Patterns, Alertness and Fatigue of Shift Nurses according to Circadian Types. Journal of Korean Biological Nursing Science. 2017;19(3):198-205. https://doi.org/10.7586/jkbns.2017.19.3.198)

---

## [Editor Report · Decision Letter 1]

22 Jun 2022

Health problems, turnover intention, and actual turnover among shift work female nurses: Analyzing data from a prospective longitudinal study

PONE-D-21-32791R1

Dear Dr. Choi-Kwon,

We’re pleased to inform you that your manuscript has been judged scientifically suitable for publication and will be formally accepted for publication once it meets all outstanding technical requirements.

Kind regards,

Mohammad Hossein Ebrahimi

Academic Editor

PLOS ONE
---

## [Editor Report · Acceptance letter]

28 Jun 2022

PONE-D-21-32791R1 

Health problems, turnover intention, and actual turnover among shift work female nurses: Analyzing data from a prospective longitudinal study 

Dear Dr. Choi-Kwon:

I'm pleased to inform you that your manuscript has been deemed suitable for publication in PLOS ONE. Congratulations! Your manuscript is now with our production department. 

Kind regards, 

on behalf of

Dr. Mohammad Hossein Ebrahimi 

Academic Editor

PLOS ONE